# Is an Exercise Program for Pediatric Cancer Patients in Palliative Care Feasible and Supportive?—A Case Series

**DOI:** 10.3390/children10020318

**Published:** 2023-02-07

**Authors:** Ronja Beller, Gabriele Gauß, Dirk Reinhardt, Miriam Götte

**Affiliations:** Department of Pediatric Hematology/Oncology, Center for Child and Adolescent Medicine, Clinic for Pediatrics III, West German Cancer Centre, University Hospital Essen, 45147 Essen, Germany

**Keywords:** terminal care, quality of life, child, adolescent, psychosocial support systems, neoplasms, physical activity, training

## Abstract

(1) Background: Growing evidence indicates benefits through exercise programs in pediatric oncology throughout the whole cancer trajectory. This should include palliative care, too. This project analyzes the feasibility of a supervised exercise program offered during hospital and home-based care for children with advanced cancer diagnoses. (2) Methods: Four children (7–13 years old) with advanced cancer diagnoses participated in this project. It consisted of supervised exercise sessions offered once a week (30–90 min), mainly home-based, but also on an in- and outpatient basis. Regular data assessments included psychological and physical capacity-related endpoints and body composition. Details and contents of exercise sessions and adverse events were recorded. (3) Results: Exercise was feasible with 73 ± 9% adherence to the minimum number of planned sessions. The exercise offer was accepted until shortly before death. Effects on fatigue, quality of life and muscular endurance were noted. Participants showed major deviations from age-specific reference values. No exercise-related adverse events occurred. (4) Conclusions: The exercise program was safe, feasible, and might have served as a supportive tool to reduce overall burden. Evaluation of exercise as usual palliative care should be assessed by further studies.

## 1. Introduction

Benefits of exercise programs have been shown in adult and pediatric oncology, such as in physical fitness, body composition, flexibility, cardiorespiratory fitness, muscle strength, health-related quality of life, fatigue and immune function [1,2,3]. Therefore, patients should have access to exercise programs during the whole cancer trajectory, including palliative care [4]. These exercise programs should be goal-oriented, playful, age-appropriate, individualized, supervised, and focus on individual strengths and weaknesses as well as preferences. In terms of content, they should consist of a mixture of strength, endurance, coordination and flexibility training [2,4,5]. Improving or maintaining quality of life, pain management, access to physical rehabilitation and physiotherapeutic measures are important in palliative care [4,5]. Exercise led by clinical exercise physiologists is not specifically included in existing palliative care guidelines. In adult patients with terminal cancer, exercise has been found to be feasible, safe, and to improve physical performance as well as quality of life and to lower fatigue [6]. It could lessen symptoms (e.g., nausea, shortness of breath or muscle wasting), too [7,8]. Limited data are available for adolescents and young adult patients [9,10]. In pediatric advanced cancer patients, data on feasibility, methods and effectiveness of exercise programs are lacking. The aim of this case series was to introduce an exercise program with home visits for children with a terminally ill cancer diagnosis. It was prepared following the CARE Guidelines (https://doi.org/10.1016/j.jclinepi.2017.04.026, accessed on 15 December 2022) (see Checklist in Appendix A).

## 2. Materials and Methods

A family that previously participated in our exercise program asked if the exercise physiologists could continue to support them and their child with advanced cancer and visit her at home. Therefore, this patient-initiated project was implemented to evaluate the feasibility of an exercise program in a palliative setting. Inclusion criteria were: (a) >3 years old with advanced cancer, (b) previous participation in the hospital exercise program, (c) resident in the area surrounding the hospital (<1 h drive). Four patients with different diagnoses participated in this project. A 13-year-old female and an 8-year-old male with relapsed alveolar rhabdomyosarcoma, a 7-year-old male with relapsed acute lymphoblastic leukemia, and a 12-year-old male with relapsed Ewing sarcoma and their legal guardians obtained written consent for participation.

The exercise program consisted of individual (1:1) sessions supervised by a qualified exercise physiologist in pediatric oncology at least once a week and lasting 30 to 90 min per session. The program was mainly home-based, but also offered on an in- and outpatient basis if patients had appointments in the hospital. It comprised a combination of strength, endurance, coordination, body awareness, and mobility training. Content was resource-oriented and related to each child’s interests as well as adapted to their respective daily condition. An example of an exercise session for the 13-year-old female is shown in Table 1.

Data assessment was scheduled at baseline (first home visit) and every 4–6 weeks depending on the condition of participants. It included (a) psychological endpoints: health-related Quality of Life (HRQoL), Multidimensional Fatigue Scale (both Pediatric Quality of Life Inventory [11], PedsQL, short form 15 and multidimensional fatigue scale) and; (b) physical capacity-related endpoints: muscular endurance/leg strength (sit-to-stand 5 repetitions), isometric maximum force/handgrip strength (Jamar Plus+ Dynamometer, Performance Health Supply, Huthwaite, UK); and (c) body composition (BIACORPUS RX 4004M, MEDI CAL, Karlsruhe, Germany). All assessments were chosen based on previous studies and recommendations for pediatric exercise oncology [12]. Adherence to training was calculated using the relation of minimum number of planned sessions (one exercise session per week) related to the total number of weeks they participated in the entire program (number of weeks from start of project until death). Basic information and contents of exercise sessions and adverse events were recorded. Results of physical capacity-related endpoints to reference values of healthy children were calculated at baseline. Values were adjusted to age- and gender-matched normative data by analyzing the percentage difference between test results and normative data.

## 3. Results

Adherence to planned exercise sessions was 70 ± 9% with a total duration of participation for 3–18 weeks, with 44% home visits and 31% inpatient and 25% outpatient sessions. The offer of the exercise program was accepted until shortly before death, with the last intervention being 10 ± 3 days prior. Mean duration of each exercise session was 61 ± 12 min. Two children were able to participate in longitudinal data assessments. The other two patients died too soon or were too weak for the assessments. Baseline data (T0) are summarized in Table 2 with anthropometric data and exercise-related parameters. Maximal isometric handgrip strength and muscular endurance/leg strength (5× sit-to-stand) were compared to reference values for healthy children. In isometric handgrip strength, patients had deviations of −5% to −77% compared to healthy children. Leg strength differed from around −90% to +19%. No exercise-related adverse events occurred during or after the exercise sessions.

Physical capacity was reduced in comparison to published age-specific reference values. In handgrip strength, participants deviated up to −77% from the reverence values. In muscular endurance/leg strength, the deviation was up to −90% (Table 2).

### 3.1. Longitudinal Results from the 13-Years-Old Female

The 13-year-old female was diagnosed with alveolar rhabdomyosarcoma of the right calf (FOXO1-FISH positive) in September 2018 with lung metastases bilaterally and lymph node metastases mediastinal. Primary therapy was according to CWS guidance, VHR group, CEVAI therapy. In January 2019, the patient underwent tumor resection in the right calf with local soft tissue reconstruction to preserve leg function. Although slight gait impairments were noticeable, the patient did not experience limitations in everyday life activities. In pulmonary function testing, FEV1 was 2.13 L/77%, and FVC was 2.13 L/68% in May 2020 (no further examinations made). In October 2020, the first relapse occurred with clinical respiratory distress, thoracic neoplasm and a new partial atelectasis of the left upper lobe as well as metastases in lymph nodes and the left lower lobe. Recurrence chemotherapy according to CWS guidance with ACCTTIVE, including a change to trabectedin, was given in February 2021due to progression with total atelectasis of the left lung. The patient showed no response, resulting in further cancer progression and discontinuation of therapy. Reduced LV function occurred with an EF of just under 50%, with a systolic LV size barely above normal (Figure 1).

She started the exercise project at the beginning of December 2020, shortly after the second ACCTTIVE cycle. She participated with 69% adherence to exercise sessions for 18 weeks of participation, with eight home visits and seven inpatient and two outpatient sessions.

Her fatigue score increased from T0 to T1/T2, resulting in fewer problems with fatigue before decreasing again. The General Fatigue subscore from T0 to T1 indicated slightly fewer problems but decreased slowly thereafter. The Sleep/Resting Fatigue subscore increased from T0 to T2, resulting in fewer problems with sleep/resting fatigue before decreasing again. The Cognitive Fatigue subscore increased and remained above baseline, indicating fewer problems. HRQoL showed a slight increase in general, especially in the Psychosocial Health subscore. The Physical Health subscore was, in total, much lower and decreased again after an increase (Figure 2A,B).

For physical capacity-related endpoints, muscular endurance/leg strength increased as indicated by a 26% shorter time to complete the sit-to-stand test (Figure 2C). These results differed from age- and gender-specific reference values by between 90% and 11%. Isometric maximum handgrip force decreased slightly over time (Figure 2D), which resulted in a maximum difference of 4% relative to age- and gender-specific reference values. Her body composition, with a BMI of 19.5 kg/m² and a fat mass between 12.7 kg (T0) and 13.4 kg (T3), was within normal age-specific reference values. Phase angle lowered (T0: 3.5°, T1: 3.9°, T2: 3.2° and T3: 3.1°) and was generally low compared to mean phase angle 5.48° ± 0.9 for females (10–13 years, BMI between 15–20 kg/m²) [15]. Body cell mass (metabolic active tissues of the body) tended to decrease (T0: 13 kg; T1: 14.1 kg; T2: 12.2 kg; T3: 11.6 kg), whereas extracellular mass (metabolic inactive) tended to increase (T0: 23.3 kg; T1: 22.6 kg; T2: 24.3 kg; T3: 24.9 kg), which resulted in the same body mass of around 49.5 kg over time.

### 3.2. Longitudinal Results from the 8-Year-Old Male

The 8-year-old male was diagnosed with alveolar rhabdomyosarcoma of the right hand (PAX3-FOXO1-Translokation) in March 2021, with lymphogenic metastases including the right axilla, bipulmonary, mediastinal and cerebral metastases. Primary therapy was according to CWS guidance for metastatic disease with extirpation cystic metastasis, radiotherapy of the primary tumor and maintenance chemotherapy with vinorelbine and cyclophosphamide. In January 2022, progression occurred under maintenance therapy with four new metastases in the cerebrum treated with relapse chemotherapy adapted from phase I study Protocol I3Y-MC-JPOCS(b). A first generalized seizure occurred in March 2022, resulting in a structural epilepsy. During further progression, a change to palliative chemotherapy according to the RIST-rNB-2011 protocol was necessary, and this was when he joined this project in April 2022. During therapy, he suffered from strong nausea, headache and suspicion of intermittent anxiety attacks. RTST was cancelled in October 2022 due to further progression, resulting in exitus letalis shortly after. In March 2022, his LV systolic function and size were normal with EF 71%, which reduced to EF 57% in July 2022. By the beginning of October 2022, he was able to join his hockey training and school in reduced hours and intensity.

His fatigue total score decreased over time, indicating higher levels of fatigue, but with missing values for T2. The Cognitive Fatigue and General Fatigue subscores decreased, but with Sleep/Rest Fatigue increasing over time. HRQoL showed a slight decrease for the total score, which was particularly noticeable in the Physical Health Summary subscore as well as at T3 in the Psychosocial Health Summary subscore (Figure 2A,B).

Muscular endurance/leg strength tended to stay nearly the same in the sit-to-stand test, which was compared to age- and gender-specific reference values of up to 42% better (Figure 1C). Isometric maximum handgrip force decreased slightly over time and was −10% to –16% below reference values (Figure 2D).

Body composition values were not measured, because the patient refused the measurement.

## 4. Discussion

This exercise program with four pediatric advanced cancer patients seems to be safe and feasible as well as gladly accepted and gratefully received. Feedback from patients and families was exclusively positive. They accepted the offer of the individual exercise program willingly and requested it when the patient’s general health allowed it. Parents reported an improvement in their children’s mood and described the sessions as the most important activity of their day. For the 13-year-old female, the appointments were a great motivation to get dressed and put makeup on. The 12-year-old male asked if the exercise program could be continued even if he moved to hospice. Families mentioned that they did not have the strength and resources to bring their child somewhere to participate in an exercise program but instead welcomed the home visits.

The need for this kind of exercise program for pediatric advanced cancer patients is underlined by major deviations from age-specific reference values. These differences were much higher compared to other studies with children at the end of acute treatment phase [16,17]. Although it was not the goal to reach reference fitness levels, better physical capacity enabled mobility and self-reliance in everyday life.

On a psychosocial and physical level, the patients seemed to have benefited from this exercise program, which could serve as a supportive tool to reduce overall burden. Personal connection and previous history in the clinical exercise program seemed to be important prerequisites for success.

Data collection was challenging due to the patients’ health conditions. Exercise professionals always respected the patients’ barriers and boundaries. If participants did not feel strong enough for testing, the usual adapted exercise program was performed instead.

Close involvement by the holistic treatment team to be informed about the patient’s current physical condition and limitations is important to conduct safe training. No adverse events occurred in the context of exercise. This is in line with the literature on exercise interventions in pediatric oncology [18] and in adult advanced cancer patients [6].

Further studies should analyze the benefits and feasibility of such particularly time-consuming interventions, and thus further expand and promote projects that can be a great resource for affected families.

## 5. Conclusions

The exercise project for these four children with advanced cancer seems to be safe and feasible, and it supported participants and their families during the enormous burden. It was helpful in this project to have regular exchanges with the treatment providers and always take the needs of the families into account. Further studies and projects in this field should be conducted.

## Figures and Tables

**Figure 1 children-10-00318-f001:**
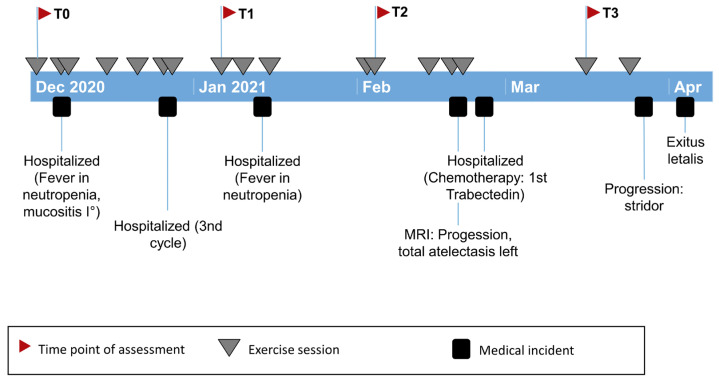
Timeline of 13-year-old female with alveolar rhabdomyosarcoma during the exercise project from December 2020 until April 2021, with four timepoints (T0–T3), marked exercise sessions (grey triangle) and medical events (black square).

**Figure 2 children-10-00318-f002:**
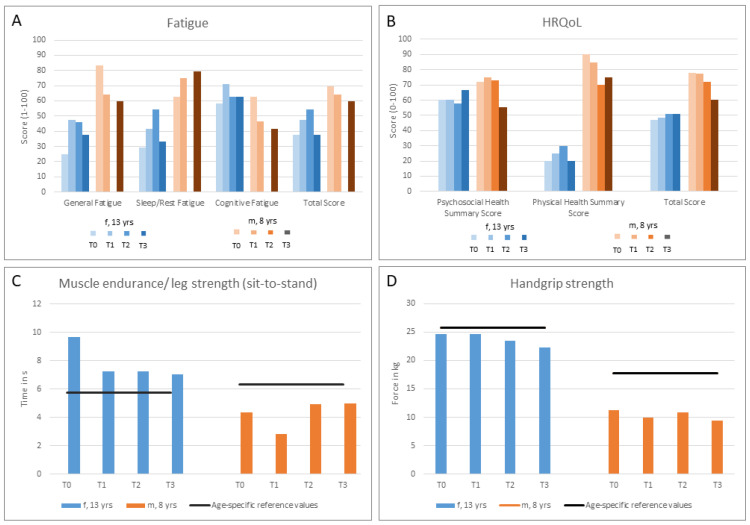
Results of psychological and physical assessments at different time points (T0–T3) for the 13-year-old female (f, 13 yrs) and the 8-year-old male (m, 8 yrs). (**A**) Subscales and total score for Multidimensional Fatigue Scale. Higher scores indicate reduced problems with fatigue. Missing values for T2 for m, 8 yrs. (**B**) Subscales and total score for Health Related Quality of Life (HRQoL). Higher scores indicate better HRQoL. (**C**) Muscular endurance/leg strength compared to age- and gender-specific reference values (5× Sit-to-stand). Lower times (s) indicate better results. (**D**): Maximal isometric handgrip strength compared to age- and gender-specific reference values. Higher results indicate more force.

**Table 1 children-10-00318-t001:** Example of an exercise session during a home visit with the 13-year-old female.

		Training in Strength, Endurance, and Motor Skills
Session objective:		Distraction from the overall medical condition, improvement of physical and psychological well-being, maintenance of mobility in everyday life
**Duration:**	**Section:**	**Content:**
10 min	Opening	Arrive, obtain information about the current state of health; set daily goal and define content of the session together.
5 min	Warm-up	Warm-up through mobilization exercises: Circle wrists and ankles, circle shoulders, spinal mobilization with cat-cow exercise, rotation of upper body with stable pelvis from right to left, hip mobilization through pendulum movements of free leg while standing on one leg.
10 min	Training in motor skills	Training in motor skills: Life Kinetics exercises with socks, e.g., throwing ball spinning top (2 pairs of socks are held in front of the body, left hand throws the pair of socks steadily and constantly 20 cm high, while the right hand throws the second pair of socks 40 cm high; variations: Change sides; instead of the 40 cm throw, the pair is thrown up, then circled with the whole hand and caught again; circled in both directions).
25 min	Strength endurance training	Strength endurance training: Playing bingo by rolling a single die, with movement tasks defined per numerical value (e.g.,: 1: 3× stand up and sit down; 2: 10 chest passes with a ball; 3: strengthening shoulders by side lifts while sitting; 4: 1 min endurance training on the stepper; 5: strengthening lower back; 6: puzzle task).
10 min	Cool down	Cool down: imaginary journey for relaxation and body awareness
3 min	Closing	Feedback on the exercise session, asking how participant feels now, goodbye
Voluntary additional exercise task		35 day sport challenge “Time to Shine”, with individual exercise tasks for each day to tick off.

**Table 2 children-10-00318-t002:** Anthropometric and physical capacity-related endpoints at baseline testing T0 compared to age- and gender-specific reference values.

Patient	BMI	Age (Years)	Gender	Difference from Reference Values					
				Handgrip Strength Left		Handgrip Strength Right		Muscular Endurance/Leg Strength	
				Absolute value (kg)	Percentage value (%)	Absolute value (kg)	Percentage value (%)	Absolute value (s)	Percentage value (%)
1	19.4	13	f	−1.2	−5.1	−1.9	−7.4	−5.1	−89.7
2	14.9	7	m	−10.8	−77.3	−10.4	−71	*	*
3	11.9	12	m	−10.4	−41.3	−15	−56.4	−2.8	−52.1
4	15.8	8	m	−7.5	−42.6	−11.9	−62.5	1.2	19.2

Note: Handgrip strength values were compared with the reference values for each single test item [13]. Muscular endurance/leg strength values were compared with unpublished reference values (n = 289) in Munich [14]. * missing data due to health conditions.

## Data Availability

The data presented in this study are available on request from the corresponding author.

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
