# Peer review of "Is an Exercise Program for Pediatric Cancer Patients in Palliative Care Feasible and Supportive?—A Case Series"

_children, 2023, doi:10.3390/children10020318_

Round 1

Reviewer 1 Report

Thank you for the opportunity to present your case report "Is an exercise program for pediatric patients in palliative care feasible and supportive? - A case series" by Ronja Beller et al.

First, I would like to express my appreciation that this topic/group of patients is being highlighted. We have seen similar improvements in mobility and quality of life in palliative care patients who were fit enough to participate in our inpatient rehabilitation program, and the answer to the original question is yes. The problem is that there are very few patients in this condition who are still strong enough for rehab, and so data collection is very difficult.

The idea of promoting physical training programs in palliative care patients should be self-evident, without having to prove the benefit by collecting data from these patients. However, it is necessary for funding. Data collection was done carefully and with consideration for the patients, so some data are missing, which is inevitable in this patient group. 

All parameters, but especially the quality of life questionnaires, showed an improvement during implementation, followed by an inevitable decrease in scores due to progression, which is a clear sign of meaningfulness in this patient group. 

Author Response

Dear Reviewer,

Thank you very much for your appreciative feedback.

We also believe that our work in this particularly vulnerable group is very important and might support patients to maintain quality of life and normality. We started the project because of a wish of one of our patients. This patient never wanted us to fuss over her and felt through the project that we did the home visits not only for her but also for science and therefore for other patients. Therefore data collection was not our main goal. And we totally agree that missing data in this patient group is inevitable. 

We are really happy that we could see the improvements in our data, too, as well as in positive feedback of the patients or their parents.

Thank you again for your supportive comments.

Reviewer 2 Report

I thank the authors for carrying out this interesting study. I think it is important to underline that the type of study carried out is not a case report but a case series. I suggest referring to the CARE guidelines for better study conduct and writing. In fact, I think it is appropriate to insert a timeline that helps the reader to better understand the process of the patients followed. I think the introduction needs to deepen some aspects such as for example the effectiveness of the therapeutic exercise on the cancer patient and the methods of delivery. Also I have a question for the authors: why was the exercise conducted by what you call an "Exercise physiologist" and not by a physiotherapist? I think the physiotherapist is the ideal figure for this type of intervention.

Best regards

Author Response

Dear Reviewer,

thank you very much for your feedback. We will answer your comments one by one in the following.

  • I think it is important to underline that the type of study carried out is not a case report but a case series.
    • We totally agree with you, but we could not chose case series as a type. So instead we choose case report, but sticked with case series in our title.
  • I suggest referring to the CARE guidelines for better study conduct and writing. In fact, I think it is appropriate to insert a timeline that helps the reader to better understand the process of the patients followed.
    • We have followed the CARE guidelines as much as possible. And decided to omit a timeline since we already have two tables and one figure. But of course we understand if our case series is easier to understand with a timeline. Therefore, we decided to add one as an exampleas well as the checklist of CARE guidelines as supplementary material. Thank you very much for this feedback.
  • I think the introduction needs to deepen some aspects such as for example the effectiveness of the therapeutic exercise on the cancer patient and the methods of delivery.
    • Thank you very much for this comment. We added some more aspects.
  • Also I have a question for the authors: why was the exercise conducted by what you call an "Exercise physiologist" and not by a physiotherapist? I think the physiotherapist is the ideal figure for this type of intervention.
    • Thank you for this question. We use the term "exercise physiologist"  as an umbrella term for all professions who are qualified to carry out exercise therapy with children and adolescents. This includes persons with training as professional movement specialists with at least three years of academic or corresponding non-academic training.
      So with this term we try to include physiotherapists. In Germany physiotherapists usually do not work scientifically. In addition, physiotherapists in Germany work with prescriptions from doctors and are limited in time. An additional exercise program for all patients in pediatric oncology is unfortunately not yet funded and supported by law. Therefore, we have sport scientists and exercise therapists who offer exercise programs financed by third-party funds and who scientifically evaluate these programs. However, in order not to exclude physiotherapists from this term sports scientist, we use the term exercise physiologist.

Again thank you very much. We value your feedback.

Nice regards.